# Alu elements in primates are preferentially lost from areas of high GC content

Elizabeth HB Hellen and John FY Brookfield

Centre for Genetics and Genomics, School of Biology, University of Nottingham,
University Park, Nottingham, UK

## ABSTRACT

The currently-accepted dogma when analysing human Alu transposable elements is that 'young' Alu elements are found in low GC regions and 'old' Alus in high GC regions. The correlation between high GC regions and high gene frequency regions make this observation particularly difficult to explain. Although a number of studies have tackled the problem, no analysis has definitively explained the reason for this trend. These observations have been made by relying on the subfamily as a proxy for age of an element. In this study, we suggest that this is a misleading assumption and instead analyse the relationship between the taxonomic distribution of an individual element and its surrounding GC environment. An analysis of 103906 Alu elements across 6 human chromosomes was carried out, using the presence of orthologous Alu elements in other primate species as a proxy for age. We show that the previously-reported effect of GC content correlating with subfamily age is not reflected by the ages of the individual elements. Instead, elements are preferentially lost from areas of high GC content over time. The correlation between GC content and subfamily may be due to a change in insertion bias in the young subfamilies. The link between Alu subfamily age and GC region was made due to an over-simplification of the data and is incorrect. We suggest that use of subfamilies as a proxy for age is inappropriate and that the analysis of ortholog presence in other primate species provides a deeper insight into the data.

## INTRODUCTION

Only a small proportion of our genome is made up of sequences that code for proteins (*Lander et al., 2001*). Transposable elements are found abundantly in non-coding DNA and the Alu family of SINE transposons accounts for approximately 10% of the total DNA in the human genome (*Cordaux & Batzer, 2009*). Several Alu subfamilies are known to be actively transposing and it is thought that a new insertion occurs approximately every 20 births in humans (*Cordaux & Batzer, 2009*). However, most Alu sequences in the genome are members of abundant, formerly active, subfamilies that are now transpositionally inert. Some Alu insertions are disease-causing (*Callinan & Batzer, 2006*; *Gallus et al., 2010*; *Boone et al., 2001*), and insertions into genes and other functional sequences are

Corresponding author
Elizabeth HB Hellen,
elizabeth.hellen@nottingham.ac.uk

typically eliminated by natural selection. However, many functional DNA sequences have been derived from Alu sequences-Alus have contributed to the control of transcription by supplying transcription factor-binding regions (*Laperriere et al., 2007*; *Polak & Domany, 2006*; *Cowley & Oakey, 2013*) and they are involved in alternative splicing (*Li et al., 2001*; *Nekrutenko & Li, 2001*), and in supplying transcriptional start sites for antisense transcripts used in gene regulation (*Conley, Miller & Jordan, 2008*). These properties, of some Alu sequences, are unlikely to be neutral in their selective effects. In principle, these could be weakly harmful and yet could have spread to fixation by drift in small ancestral populations. However, the probability of spread to fixation of a positively selected variant is greatly enhanced relative to a neutral or weakly deleterious variant, with this probability being approximately twice its selective advantage, when in the heterozygous state (*Kimura, 1962*). Were the Alu elements that have spread to fixation in human populations spread by a selective advantage that they conveyed? Polymorphisms for Alu insertions have been much studied as a tool in human population genetic inference, particularly because the absence of the element can always be identified as the ancestral state (*Batzer & Deininger, 2002*). Data from the 1000 genomes project have revealed the frequency distribution of polymorphic Alu sequences, and have shown this distribution is as expected from selective neutrality (*Stewart et al., 2011*). Almost all polymorphic Alu sites are in non-coding regions, supporting a model where Alu insertions into functional sequences are selectively harmful and rapidly eliminated, and the remaining insertions are in non-functional regions, and their spread through populations occurs by genetic drift, not selection.

However, Alu elements that were, at the time of their insertion, neutral or very weakly harmful, could have subsequently evolved functional roles, a process called "domestication". Given the abundance of Alus and other transposable element DNAs in the genome, it would be strange indeed if adaptive functions never evolved in these DNAs, since there is no reason to suppose that this DNA component is constitutionally inert to advantageous changes in its base sequences. But we do not know the proportion of Alu sequences in the genome that are now functional and selectively maintained.

The human genome project (*Lander et al., 2001*) argued strongly that the Alu sequences in our genome are mostly functional. This argument was based on the mean GC-content of DNA flanking Alu subfamilies of different ages. Alu elements from subfamilies with peak transpositional activity more than 5 million years ago tend to be located in high GC, gene-rich, areas of the genome. In contrast, elements belonging to "human-specific" subfamilies, peaking in activity within the last five million years, tend to be seen in low GC, gene-poor regions. In this analysis, the age of the subfamily was taken as a proxy for the age of insertion of the element, as has continued in further studies. All elements from 'old' subfamilies, such as AluJo and AluJb, are assumed to be old elements. However, the actual age of the elements themselves has not been calculated and, therefore, all elements in each subfamily are assumed to have approximately the same age and the same insertional behaviour and mechanisms.

The Human Genome Sequencing Consortium (*Lander et al., 2001*) argued that this difference in GC content surrounding Alu sequences older or younger than five million

years ago was due to one of three possibilities. These were, firstly, that there is a higher rate of random loss of Alus in GC-poor regions, secondly, that negative selection is acting against Alu elements in GC-poor areas and thirdly, that positive selection is acting in favour of Alu elements in GC-rich areas. The first two hypotheses are dismissed as unlikely because DNA with a low GC percentage is gene-poor and has been shown to tolerate the accumulation of other transposable elements such as LINE1. The third hypothesis was assumed to be the most likely and the conclusion was that Alu sequences confer a higher level of Darwinian fitness on the individual. It has been suggested (*Conley, Miller & Jordan, 2008*) that it is a functional role of Alu sequences in supplying antisense transcription start sites that has brought about their selective retention.

Brookfield argues that this positive selection hypothesis is itself unlikely and inconsistent with our knowledge of human population genetics (*Brookfield, 2001*). Specifically, once elements have been in the genome for five million years they will be fixed in the population, and natural selection cannot, in principle, increase their abundance. He suggests that Alu sequences are inserted into both GC-rich and -poor regions. However, since there is no specific deletion process that recognises Alu sequence ends, any deletion that removes an Alu will remove other DNA too. While Alus and their flanking DNA may be deleted from GC-poor regions with little or no harm to the organism, deletions removing Alu sequences from GC-rich DNA are more likely to be harmful, since they remove functionally important sequences in addition to the Alu, and are therefore prevented from spreading in the population by natural selection. This hypothesis implies no functional importance for the Alu sequence itself, but assumes that the insertion of an Alu into a GC-rich region is likely to confer less damage on the fitness of the organism than the subsequent, imprecise, deletion.

A further study (*Belle, Webster & Eyre-Walker, 2005*) found that Alu elements are not preferentially degraded in GC-poor regions, although the study was only carried out using a comparison of human and chimp Alu sequences. This would appear to refute the Brookfield hypothesis (*2001*), although the Brookfield hypothesis would apply only to large indels, rather than the small microdeletions which were the focus of *Belle, Webster & Eyre-Walker (2005)*.

*Costantini, Auletta & Bernardi (2012)* tackle the problem from an isochore-centric position. They conclude that the difference in distributions between young and old Alu subfamilies is explained by Alu sequences being unstable in GC-poor isochores, but stable in GC-rich isochores, although it is not entirely clear whether the comparative instability in GC-poor isochores is due to a higher rate of mutational loss or lower selective constraint.

In each of these studies, elements in a subfamily are analysed as a group, assuming that all elements in the subfamily are the same age and have the same behaviour and characteristics. We investigate the GC bias shown in Alu flanking regions by differentiating between elements within a subfamily and as such discard less data.

In this study we investigate the relationship between the flanking GC content of an Alu element and its presence in modern primate species, assuming that elements found in multiple organisms at the same chromosomal location were inserted into a

common ancestor species (*Hellen & Brookfield, 2013*). The pattern of presence in other primate species shown by Alu elements from the human genome is the result of two characteristics of the elements. The first of these characteristics is age. A specific insertion of an element that is found in the genomes of multiple primate species can be assumed to be older than one found only in the human genome. The second characteristic that these patterns represent is the conservation of the elements in non-human lineages. Rather than assuming that element copies only found in humans are the result of recent transposition events, never having existed in ancestors of the other species being compared, it could, instead, be that all element copies were present before the divergence between macaque and the apes (macaque being the species most diverged from humans in our study). The pattern of orthologous elements in modern species is then explained by the orthologous elements in other species having been deleted (or mutated to a point where they are no longer recognisable as Alu elements). Such deletion events will be the result of the combination of the underlying mutation rate for element loss in the genome, coupled with the possibility that the deleted chromosome might have been selectively spread through the population as a result of the deletion being an advantageous event conferring fitness upon the individual.

## METHODS

### Data collection

All Human (GRCh37/hg19) Alu sequences in Chromosomes 1, 2, 3, 4, 21 and 22 found on the Repbase track on the UCSC browser (*Dreszer et al., 2012*) were downloaded ($n = 345,708$). 500bp of DNA up- and down-stream of each Alu sequence was also retrieved from the UCSC browser. The GC% content of these flanking regions was calculated using bespoke perl scripts. Alu elements were divided into subfamilies and the mean GC content for each subfamily was calculated. Previous studies in the literature (*Kapitonov & Jurka, 1996*; *Lander et al., 2001*) were used to divide the Alu subfamilies into age groups: very young (Ya5, Yb9, Yb8, Yd8, Yg6, Yf4), young (Yk11, Yk4, Yc3, Yc), medium (Sx3, Sx1, Sc5, Sq2, Sg4, Sz, Sq10, Sg7, Sq4, Sc8, Sx4, Sz6, Jr4, Jr, Y, Sc, Jo) and old (FAM, Sq, FRAM, Sx, Sg, Sp, Jb). This is the order in which the elements are shown in Figs. 2 and 4.

### Determining orthologs and primate profiles

All Chimpanzee (CGSC 2.1.3/panTro3), Gorilla (gorGor3.1/gorGor3), Orangutan (WUGSC 2.0.2/ponAbe2) and Rhesus Macaque (MGSC Merged 1.0/rheMac2) Alu sequences were downloaded. The presence of orthologous elements, in the other primate species, to the human Alu elements was traced by BLASTing (*Altschul et al., 1990*) the element and separately BLASTing its flanking regions, against the elements & flanking sequences found in the four other primate genomes. The 500bp flanking regions were included in the BLAST search to ascertain synteny. Matches were only retained where at least two of the three BLAST analyses (upsteam flank, Alu, downstream flank) gave a match to the same primate Alu and flanking sequences. We have assumed that top

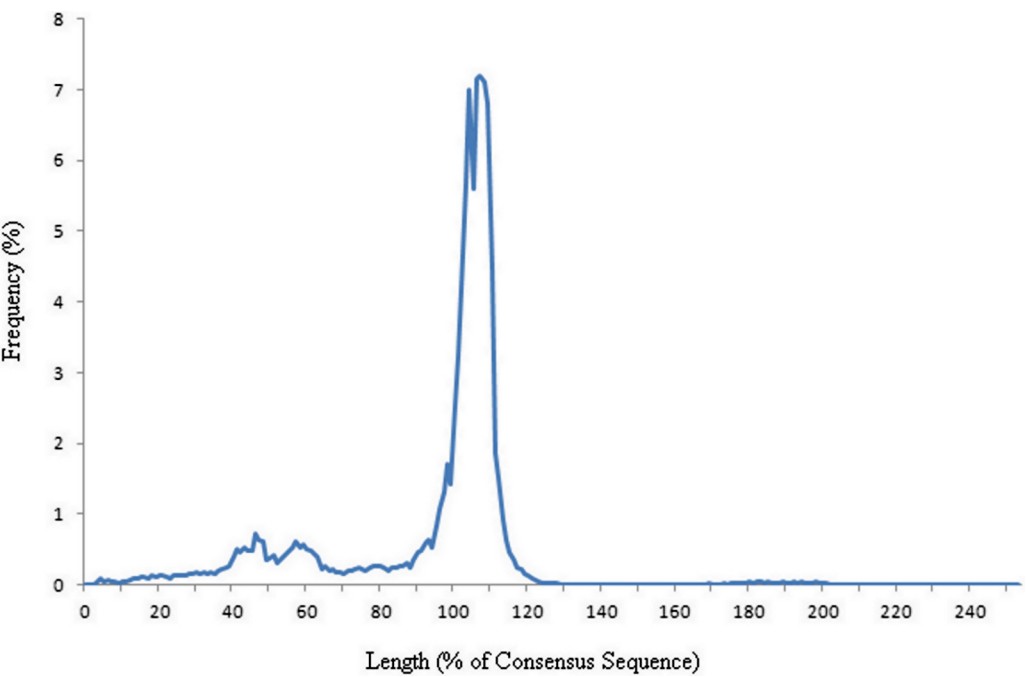

**Figure 1  Frequency graph showing the length of elements as a percentage of the consensus.** The length of each element, downloaded from UCSC, was compared to its corresponding consensus sequence. The percentage lengths are shown as a frequency graph. The majority of elements can be seen to be approximately full length.

BLAST hits which match the same Alu sub-species and have an e-value of 0.0 (which corresponds to e below 1e-200) are orthologous sequences. A frequency distribution of the length of elements compared to their consensus sequence (Fig. 1) shows that the majority of elements are 'full length' copies, however there is a smaller spike in frequency of elements between 40% and 65% length. Therefore we have decided to discard elements shorter than two-thirds of the length of the consensus sequence. This has been done to reduce the number of falsely assigned elements, shorter elements being more difficult to correctly identify due to the small number of differences in sequence between families. It is possible that there is also an issue with the validity of the GC flanking regions in fragmented elements as the flanking region may be the remains of the end of a mutated Alu element rather than representative of the environment in which the element was originally inserted. Only Alu elements with a traceable evolutionary history through the primate species have been included. Those elements without orthologs in some species where the element would be expected to be present, assuming the element was originally inserted into a common ancestor of the species in which orthologs could be found, have been removed from the dataset to reduce noise. A small fraction of these elements removed will have absences that are the result of lineage sorting, particularly for elements seen in the gorilla and human but absent in the chimpanzee. This resulted in a dataset of 103,906 elements from 34 Alu subfamilies.

Elements within a subfamily were assigned to a group depending on the primate genomes in which an ortholog could be found. Timetree (*Hedges, Dudley & Kumar, 2006*; *Kumar & Hedges, 2011*) was used to provide approximate dates (retrieved 24 April 2012) between species and confirm the phylogeny used in the analysis. The mean GC content in these groups was analysed to determine whether young and old elements within a subfamily showed the same pattern of low GC in young elements and high GC in old elements that is reported when using subfamilies as a proxy for age.

The frequency distributions of subfamily groups were compared using two-sample Kolmogorov–Smirnov tests, chosen because the test makes no assumptions about the distribution of the data. *P*-values $< 0.05$ were deemed significant.

### Analysis of whether primate profiles are related to age or conservation of elements

To distinguish between the two hypotheses explaining the presence of elements in different primate organisms (age or retention), a further BLAST search was carried out. Flanking regions of human elements were concatenated after removing the element sequence. These concatenated sequences were BLAST searched against the primate genomes in which orthologous elements could not be found. High scoring matches would indicate the likelihood of no insertion having taken place in these species.

The flanking GC% content, the element's similarity score to the sub-family consensus sequence and the length of the element was calculated for each human Alu in chromosome 1. The consensus sequences, retrieved from Repbase (*Jurka et al., 2005*), were assumed to be similar to the ancestral sequence for each family. Pearson correlation coefficients were calculated between the length of the element and the GC flanking content and between the similarity score, normalised for length, and the GC flanking content. These analyses were carried out on 29 of the subfamilies, the remaining 5 did not have a consensus sequence deposited in Repbase. If the difference in the number of species in which an Alu could be found was the result of higher levels of mutation in elements found in areas of high GC content, we would expect to see a negative correlation between GC flanking content and similarity to the consensus sequence, as a result of either deletion events or nucleotide substitutions.

## RESULTS AND DISCUSSION

### GC flanking content analysed by age of human Alu family

For each of the Alu sub-families, we first analysed the mean GC content of 500bp flanking regions on either side of each Alu element. The data were taken from 6 human chromosomes (1, 2, 3, 4, 21 and 22) and consisted of 103,906 elements, after removing those elements which could not be found in species where they would be expected, given a vertical transmission from a common ancestor of the species where orthologs could be found. The Alu subfamilies were divided by age and family into four groups: very young (Ya5, Yb9, Yb8, Yd8, Yg6, Yf4), young (Yk11, Yk4, Yc3, Yc), medium (Sx3, Sx1, Sc5, Sq2, Sg4, Sz, Sq10, Sg7, Sq4, Sc8, Sx4, Sz6, Jr4, Jr, Y, Sc, Jo) and old (FAM, Sq, FRAM, Sx, Sg, Sp, Jb) (*Kapitonov & Jurka, 1996*; *Lander et al., 2001*).
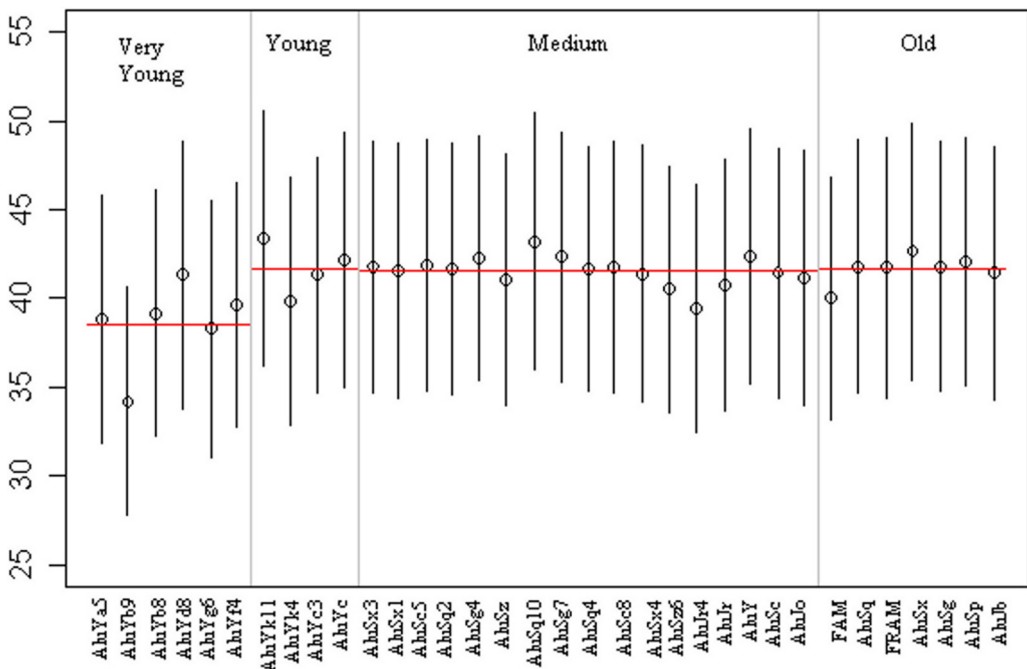

**Figure 2 The GC flanking regions of Alu subfamilies with subfamily as a proxy for age.** 34 Alu subfamilies are grouped into 4 ages: very young, young, medium, old. The mean GC flanking content of each family is plotted. Horizontal red lines represent the mean GC content of the age group, with bars showing standard deviations.

The mean GC content of flanking regions of the very young subfamily group is much lower than those in the other three groups (Fig. 2), although the standard deviation bars show the large overlap in flanking GC content that can be seen between the families. Several subfamilies characterised as 'young', 'medium' or 'old' have a lower mean GC flanking content that is more consistent with the 'very young' group, however, in general a difference can be seen between the very young group and the older age groups. The observation that lower GC flanking content is associated with very young Alu subfamilies is consistent with previous research (*Lander et al., 2001*).

## GC flanking content of elements analysed by presence in other primate genomes

The frequency distribution observed when dividing elements into groups according to their presence in primate genomes, rather than by subfamily (Fig. 3), shows a very different picture from that seen in Fig. 2. Here we see that elements only present in the human genome have a high GC flanking content and elements found in more of the primate genomes analysed have a lower GC profile. This pattern was also shown in the larger analysis which included elements where the evolutionary history of the element was less certain (e.g. orthologs could be found in Chimpanzee and Orangutan but not in Macaque), however the inclusion of these other elements also increased the level of noise, as might be expected (Fig. S1).
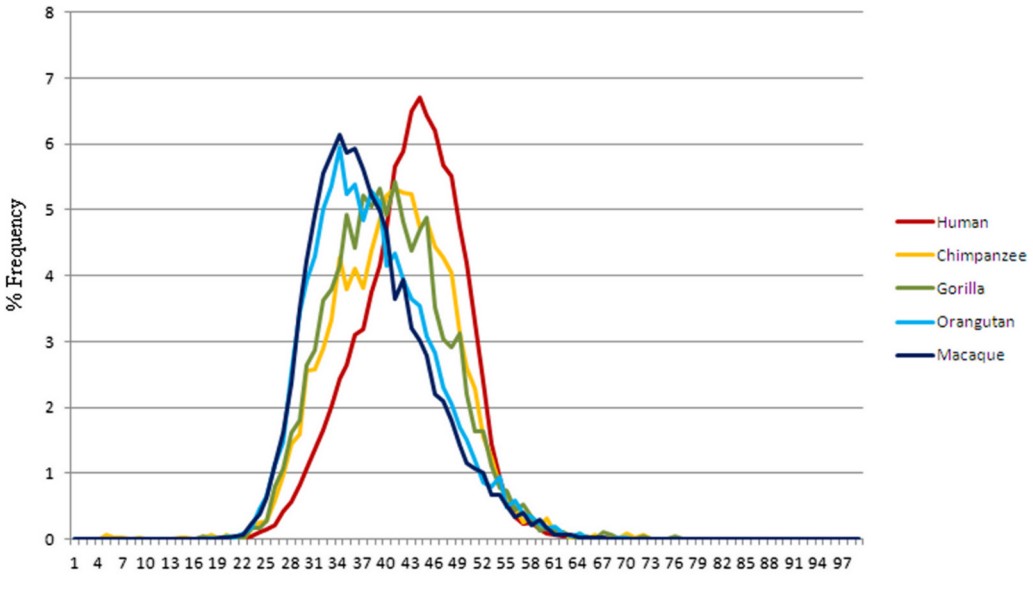

**Figure 3 The frequency distribution of Alus with presence of primate orthologs as a proxy for age.** Frequency distribution of GC content of human Alu flanking regions, categorised by the most distant primate species in which orthologs could be found. Elements are found in the labelled species and all primate species with a smaller divergence time from human, but not in the species with a larger divergence time from human e.g. a sequence labelled "Macaque" is found in all five species, a sequence labelled "Chimpanzee" is found in chimpanzees but not gorillas. GC content is calculated in 500bp up- and down-stream of the Alu element sequence in the human genome.

   If the distribution patterns shown here hold true for each subfamily and if we assume that the profile of presence of elements in primate genomes can be used as a proxy for age, then we have younger elements with higher GC values than older elements, running contrary to the data shown in Fig. 2. If the presence/absence of orthologs in other species reflects age, then the high GC content of human-specific flanking regions represents the GC bias at, or close to, the moment of insertion, while the lower GC content in DNAs flanking older elements represents preferential loss, in the lineage leading to humans, of elements from high-GC loci. Alternatively, if the presence or absence of othologous Alu elements in other genomes is not a result of differences in the age of insertion of the element, but rather of the mutational and selective forces acting on the element in other species, the data imply that Alu elements are preferentially retained in GC-poor areas, while elements have been removed from some of the other primate genomes in areas with higher GC content. If either of these explanations is true, it would lead to the conclusion that elements in areas with high GC content are preferentially lost from primate genomes. This hypothesis of a higher rate of loss of elements in areas with high GC content is the opposite of the hypotheses used in previous studies to explain the lower GC flanking content in very young Alu subfamilies (*Brookfield, 2001*; *Lander et al., 2001*; *Costantini, Auletta & Bernardi, 2012*).

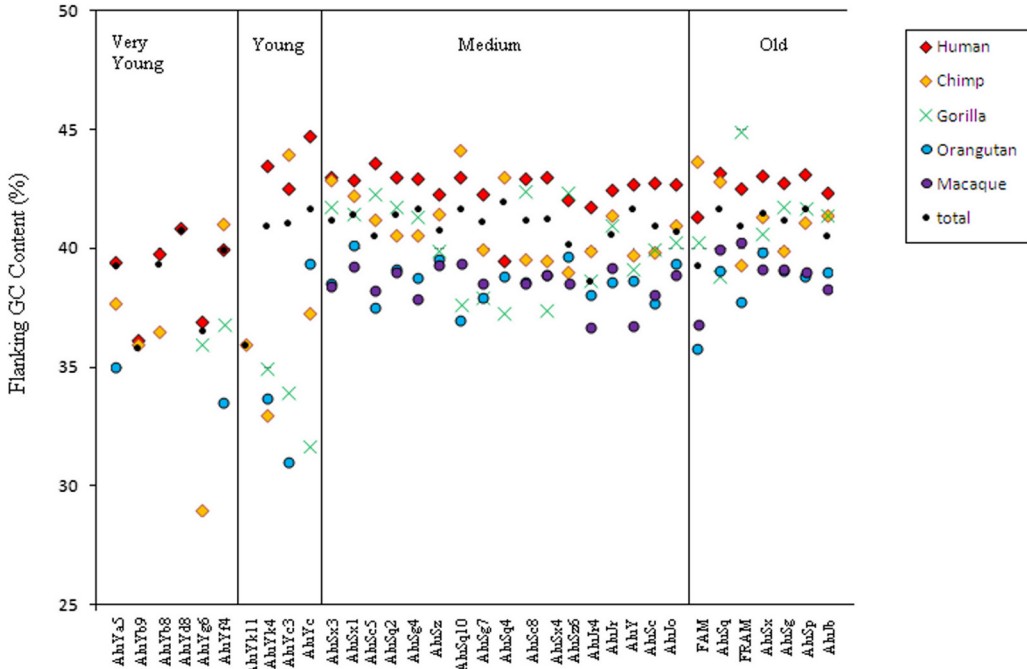

**Figure 4 The GC flanking regions of Alu elements by both primate ortholog presence and subfamily.** Mean GC content of Alu flanking regions in each subfamily, classified by presence in each of the primate species in a cumulative manner, hence 'Human' elements are found only in humans but 'Gorilla' elements are found in humans, chimpanzees and gorillas.

To analyse further this unexpected pattern of GC content in the flanking regions of Alu elements, elements were grouped by both presence in primate species and subfamily (Table S1). The data still show a trend for GC content to be higher in human-specific elements and lower in elements found in a greater number of primate species (Fig. 4). For the young subfamilies a greater range of mean values can be seen, with the older elements having much lower GC than is shown in the medium and old elements. The pattern cannot be seen in the very young elements, due to the noise arising from the shortage of elements found in the studied organisms most distantly related to humans. The orthologs to very young Alu elements are somewhat unexpected, however a careful look at these elements show that many of them are annotated as members of other subfamilies in the other primate species. It is possible that mutations in the element once in situ have changed the sequence so that it appears to be a member of one of the very young families when in an evolutionary sense, it is not (Table S2). However, the flanking regions of the human-specific, very young, Alu elements can be seen to have a much lower GC content than is observed in any of the other subfamily groupings.

For each of the subfamilies in the young, medium and old age groups, the pattern of retention of elements in primate genomes can be explained by either of the two hypotheses we have put forward. In each case, elements in high GC regions are being lost from the genome. Either old elements have lower average GC flanking regions than the younger elements because there has not been enough time for the younger elements in high GC

regions to be deleted, or the elements in a subfamily, having originally been inserted at a similar time, are more likely to be deleted from genomes when they are situated in areas of high GC content and more likely to be retained in areas of low GC content. This would result in a profile where elements found in all the primate genomes are preferentially found in lower GC regions than those only found in a few genomes. However, while the loss of Alu elements in regions of high GC content explains the data shown for the young to old subfamilies, it does not explain the low GC flanking content shown by elements from very young subfamilies.

Two-sample Kolmogorov–Smirnov tests were used to compare the total distributions of each subfamily to each of the other subfamilies, as a way of classifying the subfamilies. A group of 8 subfamilies were found with 'low' GC profiles. These subfamilies were found to have significantly different GC profiles from those with 'high' GC profiles ($p < 0.05$), but not from each other. The group consisted of five of the six very young subfamilies (Alu Yd8 did not have enough elements to show a significantly different profile from that of any other subfamily), but also contained AluYk4, AluJr4 and FAM. The inclusion of these older subfamilies in the 'low' GC profile group suggests that the GC flanking region content of Alu elements may be a characteristic of certain subfamilies and not linked to the age of the elements at all. However, a sequence analysis of the consensus sequences in each group did not show any consistent sequence differences which could be responsible for the differing profiles. It is therefore unclear what might be driving this suggested difference in insertion profile.

### Analysis of whether differences in GC profiles for elements shared by different primates are related to age or conservation of elements

Further analyses were carried out to attempt to distinguish between the two hypotheses. The two flanking regions of each of the elements found in humans, but not other species, were concatenated together, without the Alu element, and BLASTed against the other species of interest, to look for cases where those flanking sequences were not separated by an Alu. We were unable to find many clear-cut examples of positions where the flanking regions could be found with a high enough similarity to be able to assume that the element was never inserted at this position. We believe this was due to a number of reasons: a high proportion of elements being in non-coding regions where the sequence conservation was low, the divide between element and flank regions being inaccurate as many elements had low conservation in the terminal regions and, finally, a high similarity threshold.

The second analysis consisted of Pearson correlation analyses comparing the flanking GC content (as a percentage) for each element against the similarity of that element to its subfamily's consensus sequence (Fig. 5). The length of each element was also compared to the corresponding flanking GC content. The consensus sequences were assumed to be similar to the ancestral sequence. 29 analyses of each type were carried out, one for each of the subfamilies with a consensus sequence deposited in Repbase.

28 of the 29 correlations between length and GC content were found to be negative, as would be expected if elements in high GC areas were preferentially degraded through

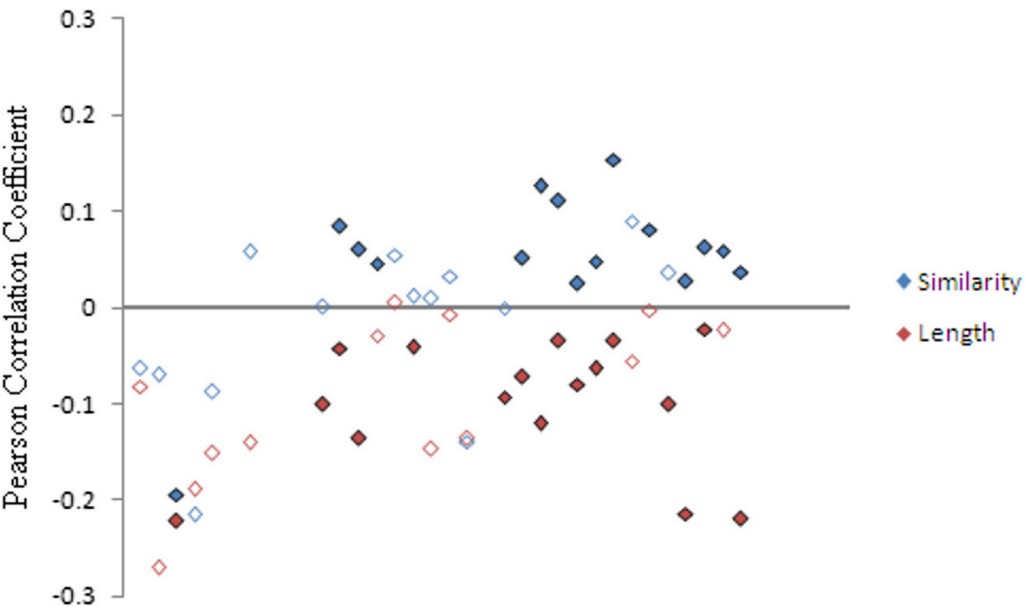

**Figure 5 Correlation between GC flanking content and similarity to consensus and between GC flanking content and length of element.** Pearson correlation coefficients were calculated, for each subfamily, between the GC flanking content and both the similarity of the element to its consensus sequence and the length of the sequence. Significant correlations are shown using filled-in points, non-significant correlations are represented by empty points.

deletion events. Sixteen of the correlations were found to be significant ($p < 0.05$). The similarity comparisons showed a positive correlation in the young, middle and old families, significantly in 14 cases. This implies that the elements in the high GC regions are closer in sequence to the consensus and have undergone fewer nucleotide mutations than those in low GC regions. However, the five very young families in the correlation analysis showed a negative correlation, although only one was significant. This difference may be due to the effects of the evolutionary processes not having had time to show in the data yet, but may also show further evidence of a different relationship between the very young elements and their genomic environment.

## CONCLUSIONS

Since the human genome project analysis was released (*Lander et al., 2001*), 10 years ago, researchers have been attempting to explain why older Alu subfamilies are found in GC-rich regions and younger Alu subfamilies are found in GC-poor regions. By analysing each element individually, rather than the mean values shown by sub-families, we have delved deeper into the issue. This closer examination of the evidence has shown that previous assumptions may have lead to misleading conclusions. Our element-specific approach clearly shows that although very young, human-specific, Alu subfamilies are found in areas of lower GC content than those of older families, this observation cannot be extrapolated to cover 'young' and 'old' individual elements. Instead, we have shown that Alu elements are preferentially removed from GC-rich areas rather than GC-poor areas.

This loss of Alu elements cannot therefore explain the presence of old Alu sub-families in GC-rich areas and young Alu sub-families in GC-poor areas.

It is not clear whether the difference between profiles of GC-content, conditional upon the presence of orthologous elements in different primate species, is differentiating elements by age of insertion or by differential retention in other species. In either case, it is clear that Alu elements are preferentially removed from areas of high GC content rather than low GC content, as was previously assumed. Some part of the loss from high-GC-content areas may be the result of selection, as some of the human specific elements may not be fixed in the population and may be subsequently lost by selection. High GC content is known to correlate with areas of high gene frequency (*Lander et al., 2001*; *Venter et al., 2001*) and as it is likely that insertions near gene regions may have negative consequences for the fitness of the host organism. However, any strong selection would have prevented insertions in high GC regions from reaching appreciable frequencies, and would thus have prevented their being included in the human-specific data, although these data could include insertions which are only very weakly deleterious. Furthermore, simple selection as a result of elements' harmful effects in high GC regions cannot explain the differences among chimpanzee, gorilla, orangutan and macaque profiles in Fig. 3.

This hypothesis of loss from high GC regions is sufficient to explain the pattern of GC bias shown within the subfamilies and to explain the patterns shown for all but the very young Alu subfamilies. The differences in GC flanking content shown by very young AluY elements appears to be due to a difference in insertion bias, a bias that is shared with three older subfamilies – AluYk4, AluJr4 and FAM. As suggested by *Belle, Webster & Eyre-Walker (2005)*, it appears as if these subfamilies have a different pattern of insertion from the remainder of the old elements. However, we could find no systematic DNA sequence differences between the 'low GC' and 'high GC' subfamily groups which would account for this differing behaviour.

### Funding
Funding for this study was provided by the BBSRC (research grant reference: BB/H009884/1). The funders had no role in study design, data collection and analysis, decision to publish, or preparation of the manuscript.

### Grant Disclosures
The following grant information was disclosed by the authors:
BBSRC: BB/H009884/1.

### Competing Interests
The Authors declare no competing interests.

## Author Contributions

- Elizabeth HB Hellen conceived and designed the experiments, performed the experiments, analyzed the data, wrote the paper.
- John FY Brookfield conceived and designed the experiments, wrote the paper.

## Supplemental Information

Supplemental information for this article can be found online at http://dx.doi.org/10.7717/peerj.78.

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
