# Peer review of "Alu elements in primates are preferentially lost from areas of high GC content"

_PeerJ, doi:10.7717/peerj.78_

## Round 0.1 · original submission · Major Revisions

Both reviewers express concerns about some different aspects of the manuscript, such as the use of truncated elements and the possibility of horizontal transfer

Reviewer 1 ·

Basic reporting

I found this paper very interesting. The results that are reported are original and intriguing. There are however a few important problems in the methodology, that need to be addressed before publication.

Experimental design

1- p. 6: "We have assumed that top BLAST hits which match the same Alu sub-species and have an e-value of 0.0 (which corresponds to e below minus 200) are orthologous sequences."

This operational definition of orthology lacks precision (because the e-value depends on the size of the database), and it is not clear whether this threshold implies that the BLAST alignment covers both the element and its flanking regions.
The authors should have measured alignment parameters (i.e. alignment length and level of sequence similarity) i) in the flanking regions and ii) in the Alu element, and then indicate clearly the minimum alignment parameters (for the element and for its flanks) that they required to infer orthology.

NB: " which corresponds to e below minus 200" should be " which corresponds to e below 1e-200".

2- P. 6: " Those elements without homologs in some species where the element would be expected to be present, assuming the element was originally inserted into a MRCA, have been removed from the dataset to reduce the likelihood of including elements which are the result of horizontal transfer."

If a given locus (an element + its flanking regions) is shared between –say - human and orangutan, but is not found in chimpanzee, then the most likely explanations are that this locus was lost in the chimp lineage or that it is missing from the chimp reference genome assembly (genome assemblies are not 100% perfect). I do not see how such cases could correspond to "horizontal transfers" (to my knowledge, the rare cases of horizontal transfers described in animals concern only transposable elements, not their flanking regions). I therefore do not understand the reason for excluding these elements from the analysis. In principle, these elements (which most likely correspond to cases of Alu losses) represent an excellent data set for the authors to test their hypothesis that Alu elements are preferentially lost in GC-rich regions. The authors should therefore include these cases in their analyses (as additional categories in Fig. 2 & 3).

I'm also surprised that the vast majority of elements do not pass this filter: only 59,676 out of 345,708 elements (i.e. 17%) are retained after this step. If 83% of loci show a taxonomic distribution that is not consistent with the phylogeny, then this implies either that the deletion rate is very high or that there are many false orthology assignments (which would of course considerably weaken the results presented here). The authors should comment on that. They should also make some controls to ensure the quality of their orthology inferences (e.g. using whole genome alignment data that are available on USCS or Ensembl).

Validity of the findings

3- Figure 3: I was surprised to see that some Alu elements from "very young" subfamilies have orthologs in Macaque. This probably corresponds to a very small fraction. The authors should provide a table with the number (and proportion) of elements having orthologs in the different primate species for each subfamily (so that the reader can see on how many elements the GC-content was computed for each point in Fig 3).

Comments for the author

I noticed a typo P. 8 : "The data were taken from 6 human chromosomes (1, 2, 3, 4, 22 and 23)".
I have only 22 autosomes in my genome... I guess I should read chromosome 21 and 22 (not 22 and 23).

Reviewer 2 ·

Basic reporting

Overall, the manuscript is well written. The authors used references quite sparsely. (The whole manuscript includes 16 references.) In particular, I suggest adding more references within the introduction. In addition, Hellen & Brookfield use commonly a review article as a reference. Please also include primary references for the cited work.

Additional labeling within the figures, in particular in Figure 1, would support a deeper understanding. For example, it would be very useful if the authors could provide the name of each subfamily instead of the categories of “very young, young, medium, and old,” and refer to the methods section. In Figures 2 and 3, I found the use of human, chimp, gorilla, orangutan, macaque potentially confusing. My suggestion is to use the typical taxonomy for the different primate groups and to include within the figure legend which species are included in the respective taxonomy term.

Experimental design

While I believe that the analyses have been thoroughly executed, I have a number of questions and concerns regarding the study design. Overall, I was excited to see the authors’ efforts to better understand the varying GC-content surrounding certain Alu subfamilies.

Based on the methods section, it is my understanding that the authors downloaded coordinates and Alu subfamily assignments for all sequences identified as Alu, including full-length and severely truncated elements. If this is the case, I imagine that the majority of Alu elements included in this study represent truncated elements. What is the fraction of truncated vs. full-length elements and what is the size distribution?

There are a number of concerns that arise from the inclusion of truncated elements:
1) The identification of the correct subfamily is more difficult and potentially not possible. In my experience, short fragments shown as a certain subfamily in the UCSC browser are identified as a (at times considerably) different subfamily if reanalysed with RepeatMasker (online as well as local installation). In context of this manuscript, I am concerned that a considerable number of the reported Alu elements are not assigned to the appropriate subfamily.
2) The presence of truncated elements may potentially introduce a GC-bias. This is because older elements are more likely to encounter insertions within the elements. If the data is not corrected for this, elements are counted more than once. This in itself may generate a bias. In addition, sections of the flanking sequence are analysed at least twice.
On page 11, the authors mention that they compared “the length of each element” to the corresponding flanking GC content. However, the authors do not provide information about their findings. Please clarify.
3) At first I was surprised that the authors could not “find many clear-cut examples of positions where the flanking regions could be found with a high enough similarity to be able to assume that the element was never inserted at this position.” I imagine that this is due to the presence of truncated elements and regions of high repeat content. Did the authors observe such a pattern? I would imagine that this analysis should be possible for full-length elements and should lead to clear-cut results in most cases.

The above described points should be addressed in the manuscript. Overall, I feel that the analysis would likely be more accurate if the authors would have focused on (close-to) full-length elements. (To reach a sufficient number of Alu elements, the authors could expand their analysis to the whole genome instead of focusing on a subset of chromosomes.) If the authors believe that truncated elements provide the same support for the analysis, different subsets should be generated to show that this is the case.

Validity of the findings

I could not find information how many Alu elements were included in subsequent analyses (e.g. how many Alu elements were retrieved per species in this study, the number of elements in each subfamily overall, human-specific elements, hominoid-specific elements per subfamily, etc.) This information should be provided (e.g. in a Table) to better follow and understand the analyses including the figures. Some of this information is especially important for the shown standard deviations in Figure 1.

Hellen & Brookfield do not use the divergence information at all. Given that it has been shown that Alu elements mutate at a neutral rate, an age estimate of Alu elements is possible. This information could be used to further support their findings. (However, severely truncated elements could cause some challenges.)

Comments for the author

In addition to my comments in the previous section I have the following (mostly minor) comments and concerns:

On page 5, data collection paragraph, first sentence the authors refer to the “Repbase” track on the UCSC browser and cite Dreszer et al. 2011. I am not aware of a Repbase track nor could I find a reference to this in the provided citation. Did the authors use the RepeatMasker track?

I noticed that the authors used Gorilla in their analysis. Did the authors find a discrepancy in the data compared to the other primates (e.g. lower number of Alu elements, different size distribution, etc.)? The motivation for my question is that I believe that quality is lower in repetitive regions compared to other primates due to the sequencing methods used.

Sometimes odd phrasing is used. For example, on page 8, second sentence in Results and Discussion section, the authors write “those elements where an unbroken lineage of orthologous elements could not be established.” Please consider rephrasing. Similarily, the last sentence in the “Determining orthologs and primate profiles” on page 6 was difficult to follow. Also, it was not clear to me what “originally inserted into a MRCA” meant. Please clarify.

The authors make the statement that they excluded certain elements “which are the result of horizontal transfer” (last sentence in the “Determining orthologs and primate profiles” on page 6). To my knowledge horizontal transfer has not been shown in primates and is thought to be very unlikely. Please clarify by either providing support for this statement or rephrase.

In the conclusions sections Hellen & Brookfield state: “…we have shown that previous studies have been looking at this issue in the wrong way.” It is true that the results presented do not support previous findings. Apart from this potentially resulting from the study design (see above), I would hope that the same points could be made more diplomatically.

On page 9, sentence starting with “If the presence/absence…” I believe it should be “GC” instead of “CG.”

---

## Round 0.2 · accepted · Accept

I believed you have successfully adressed the issues raised by the reviewers